

# Seismic noise characterisation for the Buddusò - Ala dei Sardi wind park (Sardinia, Italy) and its impact on the Einstein Telescope candidate site

Giovanni Diaferia[1], Carlo Giunchi[2], Marco Olivieri[1], Irene Molinari[1], Fabio Di Felice[3], Andrea Contu[4], Domenico D'Urso[5,4], Luca Naticchioni[6,2], Davide Rozza[7,8], Jan Harms[9], Alessandro Cardini[4], Rosario De Rosa[9,12], Matteo Di Giovanni[6,10], Valentina Mangano[5,4], Fulvio Ricci[11,6], Lucia Trozzo[12], and Carlo Murineddu[13]

[1]Istituto Nazionale di Geofisica e Vucanologia, Sezione di Bologna (Italy)
[2]Istituto Nazionale di Geofisica e Vucanologia, Sezione di Pisa (Italy)
[3]Istituto Nazionale di Geofisica e Vucanologia, Sezione di Roma (Italy)
[4]INFN Sezione di Cagliari
[5]Università degli Studi di Sassari (Italy)
[6]INFN Sezione di Roma
[7]INFN Sezione di Milano-Bicocca
[8]Università degli Studi Milano-Bicocca (Italy)
[9]GSSI Institute (Italy)
[10]Università degli Studi di Napoli Federico II (Italy)
[11]La Sapienza Università di Roma (Italy)
[12]INFN Sezione di Napoli (Italy)
[13]Renantis S.p.A (Italy)

**Correspondence:** Giovanni Diaferia (giovanni.diaferia@ingv.it)

**Abstract.** Wind turbines generate significant seismic noise and interfere with sensitive instruments, such as permanent and temporary seismic sensors installed nearby, hampering their detection capabilities. This study investigates the seismic noise emission from one of Italy's largest wind farms, consisting of 69 turbines (2 MW each), located in northeastern Sardinia. Characterizing the noise emission from this wind farm is of particular importance due to its proximity to the Italian candidate site for hosting the Einstein Telescope, the third-generation observatory for gravitational waves. We run a passive seismic experiment (WINES, 'Wind turbIne Noise assEsSment in the Italian site candidate for Einstein Telescope') using a linear array of nine broadband stations, installed at increasing distances from the wind farm. Spectral analysis, based on the retrieval of spectrograms and power spectral densities at all stations, shows a significant increase in noise amplitude when the wind farm is in operation. The reconstruction of noise polarization points out that the noise wavefield originates from a direction consistent with the wind farm's location. We recognize four dominant fixed spectral peaks at $3.4$, $5.0$, $6.8$, and $9.5$ Hz, corresponding to the modes of vibration of the wind turbine towers. While decreasing in amplitude with distance, the $3.4$ Hz peak remains detectable up to $13$ km from the nearest turbine. Assuming an amplitude decay model of the form $r^{-\alpha}$, where $r$ is the distance, we estimate a damping factor of $\alpha \sim 2$, that remains rather constant for each of the four main peaks, an observation that we relate to the good geomechanical characteristics of the local terrain, consisting of granitoid rocks. To better evaluate the possible impact



of the wind farm noise emission on the Einstein Telescope, we also analyze the seismic data from two permanent stations bordering the ET candidate site area, each equipped with both a surface and borehole sensor at approximately 250 m depth. Power spectral density analysis for the surface and borehole sensors exhibits similar results and very low noise levels. When the wind farm operates at full capacity, the borehole sensors remain unaffected by the emitted seismic noise, highlighting the significant noise suppression at depth. However, small residual spectral peaks at 3.4 Hz and between $4 - 6$ Hz remain
detectable.

## 1  Introduction

The exploitation of wind energy is of paramount importance for the transition to green energy, for the mitigation of $CO_2$ emissions. As a consequence, the installation of wind farms has been increasing in the last decades. Site selection strongly depends on wind speed and persistence over the year to maximize energy production. To minimize the impact on the local communities,
wind farms are often installed in remote areas given their low population density. Such areas, commonly characterized by low anthropogenic noise, are also ideal for installing sensitive scientific instruments (e.g. seismic stations) to minimize any disturbances on the measurements. Given such competing interests between the scientific community and the wind farm operators, adequate initiatives are necessary. For example, the state of Bavaria in Germany imposed a buffer radius around permanent seismic stations (Windenergie-Erlass – BayWEE, 2016) to preserve the detection capabilities of the local seismic network.
Several studies have targeted wind turbine installations to quantify and characterize the emitted seismic noise (Saccorotti et al., 2011; Stammler et al., 2016; Flores Estrella et al., 2017; Neuffer et al., 2017, 2019; Zieger et al., 2018; Gaßner & Ritter, 2023). The wind turbine tower vibrates with several bending and torsional modes, mainly due to the wind pressure and blade motion (Lerbs et al., 2017). Vibrational modes are effectively propagated to the ground and translate into seismic noise through surface waves. The noise generated by wind turbines has been observed for distances greater than 10 km (Schofield, 2001; Saccorotti
et al., 2011) and, while all studies agree on a typical exponential decay of the seismic noise with distance, the damping factor shows a large variability (Stammler et al., 2016; Flores Estrella et al., 2017; Zieger et al., 2018; Neuffer et al., 2017, 2019). This suggests that the geological characteristics of the site, together with the local topography and spatial layout of the installed wind turbines (Lerbs et al., 2017; Limberger et al., 2022), have a large contribution in controlling the propagation and damping of the emitted seismic wavefield. Therefore, the impact of the noise produced by an installed (or perspective) wind farm is
strongly site-specific and must be carefully evaluated with dedicated geophysical studies.

### 1.1  The Buddusò-Ala dei Sardi wind park (Sardinia, Italy) and the Einstein Telescope candidate site

In the northeastern part of Sardinia island (Italy), a large wind farm (Buddusò - Ala dei Sardi Wind Park, hereinafter referred as BAS) has been installed in a area of about $40$ km$^2$ in the municipality of Buddusò (see Fig. 1), characterized by persistent windy conditions and low anthropization. The wind farm consists of 69, 58 m tall, closely spaced, 2 MW wind turbines
(Enercon E70/2000) for a total installed power of 138 MW(https://renantis.com/it/produzione-e-stoccaggio-di-energia/). This wind farm merits the scientific attention for two reasons. First, the BAS wind park is the largest in Italy, settled on a roughly



homogeneous crystalline terrain, in one of the most seismically quiet region of the world in the frequency band $1 - 10$ Hz (commonly dominated by anthropogenic noise). Such peculiarities can lead to a more detailed characterization of the noise generated by a wind park and its propagation. Secondly, the BAS wind farm is about 13 km NW from the candidate area

for hosting the 'Einstein Telescope' (ET), the third-generation gravitational wave observatory (Punturo et al., 2010). ET will consist of a set of $10 - 15$ km long laser interferometers installed in an underground research infrastructure at $200 - 300$ m depth on average for optimal suppression of seismic, acoustic, and electromagnetic noise (Hutt et al., 2017). Given its particular and unique configuration, ET is expected to provide a 10-fold increase (ET Science Team, 2020) in the detection of cosmological events that generate gravitational waves, by pushing frequency range of such phenomena down to 1 Hz (as opposed more

than 10 Hz in current detectors such as Virgo and LIGO). Such improvement will allow to observe the Universe in its early stage, before the emergence of stars, improving our understanding of the origin and evolution of the Universe, and its relation with dark energy and dark matter. The almost absent natural seismicity, low seismic hazard (Giardini et al., 2013), and its low population density make the Sardinian site a very good candidate to satisfy the requirements for ET installation and operation (Naticchioni et al., 2014, 2020; Di Giovanni et al., 2021). In fact, the long-term average of seismic noise at this site was

shown to be extremely low, approaching the Peterson's New Low Noise Model (NLNM, Peterson (1993)) for a wide range of frequencies, with a minimal diurnal and annual variation (Di Giovanni et al., 2023). In the context of the ET installation, the potential noise contribution of the nearby BAS wind park raises particular concern because the target frequency range of the ET interferometers ($1 - 10$ Hz) overlaps with the typical frequencies excited by wind parks. This work aims at characterizing the seismic noise produced by the BAS wind park, using the ground motion recorded by a temporary, linear array of nine seismic

stations that cover the distance between the BAS wind park and the ET candidate site (Fig. 1). We compute spectrograms and power spectral densities (PSDs) in order to analyze the spectral content and its relation to wind speed and mean blade rotation rate (BRR, measured in rpm) at the the wind park. Through polarization analysis, we estimate the direction of the incoming wavefield and also derive a decay law for the noise amplitude at different frequencies and for different ranges of BRR. We then provide a discussion of the results in the context of the existing literature and the possible impact on the ET site.

## 2  Data and Method

The WINES ('Wind turbIne Noise assEsSment in the Italian site candidate for Einstein Telescope') experiment consisted of a linear array of nine seismic stations (named WP[1-9]) that covered the distance between the BAS wind park and the station P2, bordering the ET candidate area (Fig. 1). The station spacing was variable (due to some terrain inadequacy and inaccessibility), ranging from 600 to 3,000 m, for a total array length of $\sim$15 km. Station WP1 was installed 10 m NW from one of the turbines,

while all other stations were positioned at increasing distances, being WP9 the farthest from the BAS wind park. The sensors consisted of Trillium Compact 20 s broadband seismmometers, except for WP1 which was equipped with a Trillium Compact 120 s. The sampling rate was set to 100 Hz for all stations. The seismic recording was about eight weeks long, from 08/03 until 30/04/2023. We integrated the seismic recordings with those from the permanent stations P2 and P3 (Fig. 1) to spatially extend the observation coverage within the ET candidate area. These stations, installed in 2021 by INFN (Istituto Nazionale di





| wind speed | | blade rotation rate (BRR) | |
| --- | --- | --- | --- |
| range (m s$^{-1}$) | % of occurrence | range (rpm) | % of occurrence |
| $0-5$ | 44.2 | $0-3$ | 1.7 |
| $5-10$ | 29.2 | $3-5$ | 2.1 |
| $10-15$ | 17.8 | $5-10$ | 29.4 |
| $15-25$ | 8.9 | $10-20$ | 66.8 |

**Table 1.** Selected classes and % of occurrence for the recorded wind speed and blade rotation rate (BRR) during the WINES experiment

Fisica Nucleare), are equipped with a pair of surface (named P2.00 and P3.00, respectively) and borehole (named P2.01 and P3.01, respectively) seismometers placed at similar depth (264 m for the case of P2 and 252 m for P3). In this study, all the data from the WINES array and the stations P2 and P3 undergo the same pre-processing steps, being demeaning, detrending, and deconvolution from the instrument response.

    The seismic records are complemented with the data provided by the Renantis company, including the blade rotation rate
of each turbine, as well as wind speed and direction, measured by a high-quality meteorological station, named 'Met Mast' and located approximately at the center of the wind farm. Both BRR and wind conditions were sampled every 10 minutes. According to the statistics on the recorded direction and strength of the wind (see Fig. 2 panel a), conditions of absent or light wind ($< 3$ m s$^{-1}$) were particularly infrequent. On the contrary, wind was persistent with velocities commonly above $5-10$ m s$^{-1}$, with a dominant incoming direction between SW and NW. As shown 2 panel b), there is a direct correlation between
the recorded wind speed (blue curve) and the mean BRR across the 69 turbines (green curve). For convenience, we subdivide the recorded wind speed and BRR into four arbitrary, partially overlapping, classes that cover the min-max excursion for each dataset. For wind speed we choose $0-5$ m s$^{-1}$, $5-10$ m s$^{-1}$, $10-15$ m s$^{-1}$. While for the BRR the classes are $0-3$ rpm, $3-5$ rpm, $5-10$ rpm, and $10-25$. The classes and their percentage of occurrence are given in Table 1. During the WINES experiment, the highest range of blade rotation ($10-25$ rpm) is reached $\sim 67\%$ of the time, even though high to very-high
wind speed ($10-25$ m s$^{-1}$) is reached about a quarter of the time ($\sim 27\%$). Moreover, the wind speed in the low to medium range ($0-10$ m s$^{-1}$) is by far the most common condition ($73\%$). This implies that high blade rotation rates are reached even for relatively low wind speeds.

## 2.1    Noise polarization

Polarization analysis allows us to estimate the direction (i.e. the backazimuth) from which the seismic signal originates with
respect to a recording station, i.e. the backazimuth. Based on the evidence that most of the noise emitted from wind farms propagates in the form of Rayleigh waves (Westwood & Style, 2017), we derive the direction of the incoming noise along the array by exploiting the elliptical and retrograde particle motion of Rayleigh wave. Given a source with unknown backazimuth $\theta$, the vertical ($z(t)$) and radial components $r_\theta(t)$ shows a (ideal) $90°$ phase shift (Claerbout, 1976) which can be compensated





for by taking the Hilbert transform of the radial component $H[r_\theta(t)]$(Stachnik et al., 2012; Ensing & Wijk, 2019; Magrini et al., 2020). To determine the backazimuth $\theta$ we iterate the rotation of the horizontal components by $\theta_i$ and calculate $H[r_{\theta i}(t)]$, in order to find the value $\theta_i$ for which the cross-correlation with the vertical component $z(t)$ is maximized:

$$CC_\theta = max \left[ \int\limits_{t_0}^{t_0+T} z(t)r_{\theta i}(t+\tau)dt \right] \tag{1}$$

Here we perform such backazimuth estimation on data chunks of 10 minutes, iterating with $2°$ steps over the full circle.

## 2.2 Power spectral density (PSD)

We evaluate the spectral content of the recorded data through the estimation of the power spectral density (PSD), which is the Fourier transform of the time-averaged signal autocorrelation $P(\tau)$ (Aki & Richards, 2002):

$$P(\omega) = \int\limits_{-\infty}^{\infty} P(\tau)e^{iwt}d\tau \tag{2}$$

where $\omega$ is frequency and $\tau$ is the time integration variable. First, we divide seismic recording into 10-minute long data chunks (which is the same sampling interval of the BRR and wind speed time series). For each chunk, the PSD is estimated with Welch's method (Welch, 1967), using $50\%$ overlapping windows of 12800 samples to assure a good resolution for high-frequency signals. PSD amplitude is given in terms of m s$^{-1}$ per squared unit of frequency (Hz) and is averaged across all windows.

## 2.3 Estimate of damping factor

The cylindrical nature of a surface wave implies that the amplitude damping due to geometrical spreading at a distance $r$ is proportional to $r^{-\alpha}$ with $\alpha = 1/2$ Novotny (1999). However, a surface wave is damped with a higher factor $\alpha$, due to the attenuation and scattering that depends on the local geology and soil condition Stammler et al. (2016); Zieger et al. (2018). We empirically estimate the damping factor $\alpha$ by tracking the amplitude of spectral peaks along the array and for different ranges of BRR, to evaluate the consistency of the decay rate with respect to the operational regime of the wind farm. To properly evaluate the decay rate in a scenario of multiple, aerially scattered noise sources, we compensate for the simultaneous contribution of all the 69 wind turbines. It can be assumed that the turbines act as quasi-random noise sources that add in quadrature rather than in phase Schofield (2001); Saccorotti et al. (2011); Neuffer et al. (2019), i.e. $N$ equal turbines produces $N^{1/2}$ times the noise of a single turbine. Thus, we divide the amplitude of each spectral peak by $N^{-1/2}$, being $N$ the total number of turbines within a certain threshold radius, that we set at 15 km. For each station along the array, the average station-turbine distance is calculated in terms of a harmonic mean $r_H = \frac{n}{\sum_{i=1}^{n}\frac{1}{r_i}}$, to underweight the contribution of isolated turbines (Neuffer et al., 2017). Finally, with a non-linear least square method (Morè, 2006) we derive the optimal value of the decay rate $\alpha$ (and its



uncertainty $\sigma$), assuming an exponential decay law of the type $A \sim r_H^{-\alpha}$, where $A$ is the amplitude of the frequency peak, $r_H$ is the harmonic average station-turbine distance.

## 3 Results

For a first appraisal of the spectral characteristics of the recorded signal across the array, we compute the spectrograms for
a total of 10 days (from 08/03/23 until 18/03/23), covering the $1-20$ Hz frequency range. In Fig. 3 we display, for stations WP1, WP3, and WP9, the resulting spectrogram paired with the seismogram for the vertical component (velocity). Panel b of the same Figure shows the recorded wind speed at the 'Met Mast' meteorological station (considered representative of the wind speed velocities across the BAS wind park). The covered period includes intervals of calm and very light wind (e.g. days 13-14/03/23) as well as days with strong wind, reaching 20 m s$^{-1}$ (see periods 10-12/03/23 and 14-16/03/23). Station
WP1 shows, both in the seismogram and the spectrogram, a high degree of noise contamination in the entire $1-20Hz$ range during the intervals of sustained wind. Well-defined, narrow-banded, monochromatic signals can be recognized at several fixed frequencies, which persist across the selected time interval, except for the few calm days (e.g. 13-14/03/23). The amplitude of the observed monochromatic signals appears to be modulated by the wind speed, thus increasing with increasing wind speed and vice versa. The spectrogram for WP3, about 2.5 km distant from WP1, shows a lower amplitude of the noise signal (note
the change in the colormap scale with respect to WP1). Here, we note again that seismic noise contaminates the entire $1-20Hz$ band, but is predominantly confined in the same narrow-banded, monochromatic signals as in WP1. The signals above 10 Hz appear to be strongly damped with respect to WP1, suggesting a higher damping effect with distance in this frequency range. Lastly, station WP9 (13 km away from WP1) does not show the narrow-band signals observed in WP1 and WP3. However, WP9 has a stronger noise contamination than WP3 (note the same colormap scale), likely caused by particular conditions of
the local site.

### 3.1 Estimation of noise source direction

We select one day (11/03/2023) that falls in the $10-25$ m s$^{-1}$ wind speed class) and is characterized by a high level of seismic noise, as shown in the spectrograms of WP1 and WP3 in Fig. 3. We estimate the direction of the incoming noise using the method described in Section 2.1. We tried different bandpass filters, $1-10$ Hz, $2-6$ Hz, $4-7$ Hz, and $5-10$ Hz
that comprise the several of the narrow bands observed in Fig. 3. Only the $4-7$ Hz band pass allowed the recovery of the noise source backazimuth with a sufficiently high cross-correlation value (0.5) between the vertical and radial components of the Rayleigh wave. In Fig. 4 we provide an example of the polarization analysis for station WP3. The vertical and E-W components are plotted in panels a and b, respectively, while panel c shows the value of cross-correlation between the vertical and radial components as a function of the backazimuth, calculated in each 10-minute long data chunk on the x-axis. The
maxima of cross-correlation indicate a stable noise source with with an average backzimuth of $322 \pm 4.8°$. The maxima of cross-correlation reaches a value of 0.8, indicating a well constrained estimate of the source origin. In Fig. 5 panel a, we show on map the reconstructed backazimuth (blue lines) for stations WP1, WP3, and WP4. At WP1, with a cross-correlation





value between $0.5$ and $0.6$ (Fig. 5 panel b), a noise source is recovered at an average backazimuth of $110 \pm 4.4°$ which is compatible with the position of the nearby wind turbine at a few tens of meter. Lastly, WP4 shows a reconstructed backazimuth

that averages to $314 \pm 4.1°$, similar to the nearby WP3, but with lower (between $0.5 - 0.6$) cross-correlation values. For the other stations of the array, the polarization analysis suffered from weaker constraints, as indicated by cross-correlation values constantly lower than the threshold set at $0.5$. This can be explained by the loss in signal coherence when distant and sparse sources generate a diffuse wavefield, lacking a predominant noise source direction to be estimated.

### 3.2   Spectral characterization of seismic noise

To better evaluate the spectral characteristics of the noise recorded during the WINES experiment, we compute the PSDs for the entire array, covering the whole recording period. We employ the method described in Section 2.2, focusing solely on the $1 - 10$ Hz frequency range. This choice stems from the observation that: i) the spectrograms (see Fig. 3) show strong damping for signals above $10$ Hz already at $2.5$ km from the wind farm (see Fig. 3), ii) $1 - 10$ Hz will be the operational frequency range of the ET interferometers. We create four subsets of PSDs, corresponding to different classes of BRR at the BAS wind farm

(see Tab. 1). The resulting PSDs are displayed in Fig. 6 for the stations WP[1-9], and for the sensors P2.00 and P3.00. In the $0 - 3$ rpm range (Fig. 6 panel a), all stations show a comparable noise level, except for WP1 which shows the highest noise contamination and distinct frequency peaks. The main ones are centered at $3.4$, $5.0$, $6.8$, and $9.5$ Hz and correspond to those observed in the spectrograms (see Fig. 3). In the $3 - 5$ and $5 - 10$ rpm ranges (Fig. 6 panel b-c) we observe a general increase of the noise level at all stations, except for P2.00 and P3.00. All the frequency peaks already observed at WP1 appear also at

the farther stations. When BRR is between $10 - 25$ rpm (Fig. 6 panel d) we observe the highest degree of noise contamination at all stations. and all the frequency peaks at $3.4$, $5.0$, $6.8$, and $9.5$ Hz become visible at almost all stations. Moreover, a clear decrease in noise magnitude becomes appreciable when moving farther from the wind park. The peak at $3.4$ Hz is visible up to WP9, $13$ km away from the closest turbine. P2.00 and P3.00, that have a stable level of noise amplitude until $10$ rpm, with higher BRR (Fig. 6 panel d) show a small increase in spectral amplitudes in the $4 - 10$ Hz range.

### 3.3   Amplitude decay with distance

By leveraging the linear layout of the WINES array, we estimate the amplitude decay with distance and its dependency on frequency. We use the method presented in section 2.3, applied to the amplitude of the spectral peaks at $3.4$, $5.0$, $6.8$, and $9.5$ Hz tracked along the seismic array. From our analysis, we exclude station WP6 due to its relatively higher level of noise (see Fig. 6, probably due to unfavorable conditions of installation and/or local source of noise. In Fig. 7 panel a-c we show the

results of the analysis of amplitude decay. For the case of BRR in the $0 - 3$ rpm range, Fig. 7 panel a) the amplitudes of the PSD peaks are rather small and do not show a clear variation with distance. The amplitudes of the frequency peak at $3.4$ Hz represent the only exception and can be fitted with $\alpha = 1.97 \pm 0.96$. In the range $3 - 5$ rpm (Fig. 7 panel b), also the peaks at $5.0$ and $6.8$ Hz show an exponential decay with distance, with $\alpha$ now ranging from $2.2 \pm 0.2$ (at $3.4$ Hz) to $2.8 \pm 0.9$ (at $6.8$ Hz). For BRR in the $5 - 10$ rpm range, all frequency peaks closely follow a well-behaved exponential decay, with a slight increase

of $\alpha$ for increasing frequencies (Fig. 7 panel c). Lastly, the case of the wind park running at full capacity ($10 - 25$ rpm, Fig.





7 panel d) provides more reliable amplitude data along the array, resulting in a better constraint of the damping factor $\alpha$, thus suffering from lower uncertainties overall. A statistical comparison (t-test with p-value=0.05) between the values of $\alpha$ retrieved for all frequencies shows that those obtained at 5.0, 6.8 and 9.4 Hz are significantly the same, with an average value of 1.96 across the three frequencies. On the contrary, the damping factor for the 3.4 Hz peak ($\alpha = 1.7 \pm 0.0$) is statistically lower than

the others. We show in Fig. 7 panel d) the number of turbines within 15 km from each station along the array and their average harmonic distance from the turbines. Stations WP[1-7] are within 15 km from each of the 69 turbines, resulting in a harmonic mean distance between $\sim 1$ km (WP1) and $\sim 11$ km (WP7) range, while for WP9 the mean harmonic mean distance is $\sim 14$ km, from 12 turbines.

## 4   Discussion

In this study we characterize the noise emission from a large wind farm in NE Sardinia (Italy), using the recordings from an array of nine broadband seismic stations deployed for eight weeks in combination with two permanent stations located in the vicinity. For the closest stations to the wind park (WP1, WP3, and WP4), the reconstruction of noise polarization in the frequency range $4 - 7$ Hz indicates a source backazimuth compatible with the position of the closest turbines, confirming that the main source of noise in the selected frequency range is the wind park itself. Possible explanations for the inability to

estimate the noise direction in other frequency bands are i) the complexity of the recorded time series due to the superposition of several signals when the bandpass filter is too wide (e.g. $1 - 10$ Hz) and/or ii) the overall poor coherence of the seismic wavefield at high frequency (e.g. $5 - 10$ Hz), complicated by the simultaneous emission of multiple noise source and by the local seismic velocity structure and topography. Spectrograms computed for a representative selection of ten recording days (Fig. 3) show a strong noise contamination, with the highest noise amplitudes that become evident when wind speed exceeds

10 m s$^{-1}$. The closest analyzed stations (WP1 and WP3) show a predominance of seismic noise in the form of narrow-banded signals at fixed frequencies, only disappearing in case of almost absent wind, such as late 13/03/2023 and early 16/03/23 (see 3 panel b). Narrow-banded signals constitute a typical spectral feature (Nagel et al., 2021) that has been extensively observed in the vicinity of wind turbines (Saccorotti et al., 2011; Stammler et al., 2016; Flores Estrella et al., 2017; Neuffer et al., 2017, 2019, 2021; Zieger et al., 2018; Gaßner & Ritter, 2023). Commonly, the signals components above 20 Hz have a central

frequency which is multiple of the (variable) BRR (Nagel et al., 2021; Neuffer et al., 2021; Gaßner & Ritter, 2023). Contrarily, the components below 20 Hz are typically found at fixed frequencies, with variable amplitude depending on BRR. These components correspond to the several monochromatic signals emerging in the spectrograms at WP1 and WP3 (Fig. 3), related to a mixture of fundamental and higher vibration modes (both oscillatory and torsional) of the tower structure (Nagel et al., 2019; Lerbs et al., 2017; Zieger et al., 2020), resulting from a complex pattern of tower resonance and blade motion. The

analysis of the PSDs provides a complete overview of the spectral imprint of the wind farm. In the $1 - 10$ Hz range, which is the interval of our interest, we recognize four main frequency peaks at 3.4, 5.0, 6.8, and 9.5 Hz. The variation in BRR does not affect their central frequencies but clearly modulates their amplitudes. When the wind farm runs at medium to low regime ($< 10$ rpm, 33% of the time), all spectral peaks are visible in the WP1 spectrum and the signal of 3.4 Hz is visible up to WP7,



which is 7.8 km away from the closest turbine. With the BAS running at full regime ($10-25$ rpm corresponding to the $\sim 67\%$ of the examined time period), the signal at $3.4$ Hz is the strongest, approaches the Peterson's New High Noise Model (NHNM) at WP1 and becomes visible at the farthest station (WP9), 13 km away from the closest turbine. Such distances fall within the ranges that are commonly found in the literature. As an example, Saccorotti et al. (2011) showed that the frequency peak at $1.7$ Hz persists up to 11 km at the seismic station near the VIRGO Gravitational Wave Observatory in Italy (Caron et al., 1997). Schofield (2001) tracked the $4.3$ Hz signal generated by a wind park up to 18 km. However, also shorter distances ($< 5$ km) are

found (Neuffer et al. (2019) and Zieger et al. (2018), demonstrating that such discrepancies reflect the wide range of variability of the local geological condition, type of installed wind turbine, areal arrangement, and type of soil foundation. Moreover, turbine height (generally called *hub height* is the main factor affecting the main emitted frequencies. A taller hub vibrates with lower frequencies Neuffer et al. (2017) generating a low-frequency noise that can travel longer distances due to the usually lower damping. We estimate the rate of amplitude decay at different frequencies and ranges of BRR, fitting an empirical power

law $r^{-\alpha}$, being $r$ the distance and $\alpha$ the damping factor. Overall the damping factor shows small variability with respect to BRR, with a value averaging $\sim 2$, resulting in a twofold implication. First, it confirms that the method employed for estimating $\alpha$ based on the scaling of the spectral amplitudes on the assumption of in-quadrature noise sources correctly accounts for the simultaneous contribution of multiple turbines. Secondly, it confirms that $\alpha$ is thus solely controlled by the local site conditions. If we focus on the range $10-25$ rpm (in which the curve fitting is affected by the lowest uncertainties), $\alpha$ does not statistically

differ for the peaks at $5.0$, $6.8$, and $9.5$ Hz. Only at $3.4$ Hz, the damping factor $\alpha$ is statistically different ($1.7 \pm 0.0$) from the value obtained at the higher frequencies. However, the range of variability of the damping factor $\alpha$ is rather limited, implying that low as well as high frequencies undergo almost the same seismic damping. This evidence can be explained by the specific characteristics of the local terrain consisting of a compact, crystalline, Palaeozoic basement with quartzite, orthogneiss, and granitoid rocks (Carmignani et al., 2012) with good geomechanical characteristics. A similar, frequency-independent, damping

factor ($\alpha \sim 2.6$) is found in Neuffer et al. (2017) for the entire $1-10$ Hz band. Contrarily, a dependency of $\alpha$ with frequency is observed in Neuffer et al. (2019), ranging from $\sim 2.3$ at around 3 Hz to $\sim 5$ around 6 Hz, implying a much stronger attenuation of higher frequencies with distance. A similar frequency-dependent behavior is also observed in Lerbs et al. (2017), for an area with loess and other unconsolidated sediments. Across the literature, the discrepancies in the retrieved values of $\alpha$ and its possible dependency on frequency can be primarily explained by the large variability in the geological characteristics of

the site examined. Moreover, topography can also play a secondary but nonnegligible role in affecting the amplitude decay with distance. In fact, the numerical modeling in Limberger et al. (2022) shows amplification and reduction in peak ground velocity (PGV) in the order of $\pm 30\%$ even in the case of a mildly rough, hilly environment. Given the complex interplay of both topography and subsurface geology, in addition to the wide range of possible types, amount, and spatial configurations of turbines in a wind farm, any generalizations on the characteristics of the emitted noise and its propagation should be avoided.

However, to ensure that the variability of the observed amplitude damping is due to the local characteristics rather than the method employed, a homogenized strategy for the decay estimation should be sought, allowing a better comparison between different studies and, thus, site conditions. As an example, in Zieger et al. (2018) the damping factor is substantially lower ($\alpha < 1$) than in this study, in Neuffer et al. (2017) and Neuffer et al. (2019). The occurrence of such a low value, despite the



presence of unconsolidated Cenozoic (thus highly damping) sediments, could be explained by the absence of an appropriate

scaling of noise amplitude that overlooks the simultaneous noise emission of multiple wind turbines.

## 4.1    BAS wind park and the possible impact at the Einstein Telescope candidate site

In the condition of a medium-high operational regime at the wind park, the spectrum at WP9 (13 km apart from the wind park)
shows a clear signal at $3.4$ Hz. Such evidence demonstrates that the ground vibrations generated by wind parks can travel at
long distances and with limited damping as a consequence of the specific geomechanical characteristics of the local terrain. The

stations P2 and P3, equipped with both surface and boreholes sensors (see 2), allow a direct evaluation of the possible impact of
the BAS wind park even at depth. Fig. 8 shows the PSDs for stations P2 and P3, for different ranges of BRR regime, referred to
the surface (solid line) and borehole sensor (dashed line). If BRR $< 10$ rpm (Fig. 8 panel a-c) the surface and borehole spectra
overlap at both stations, close to the Peterson's NLNM. Only for BRR$> 10$ rpm (Fig. 8 panel d) the surface spectra show a
higher noise amplitude, surpassing the level of $10^{-9}$ m s$^{-1}$ Hz$^{1/2}$) in the entire $3-10$ Hz range, while the borehole spectra

remain unaffected. For lower frequencies ($< 2$ Hz), the borehole and surface spectra still overlap, implying that the borehole
installation provides no suppression of seismic noise in this frequency range. The persistence of such frequencies at depth can
be explained by the high shear wave velocities associated with the crystalline basement in the area, causing rather large, high
penetrating wavelengths $\lambda$ for frequencies lower than $2-3$ Hz (e.g. $\lambda = 1000$ m assuming $V_S = 2000$ m s$^{-1}$ at 2 Hz). This is
supported by the numerical modelling in Limberger et al. (2023), showing that the wavefield around 1 Hz undergoes almost

no damping even at larger depths (e.g. 600 m) for high shear wave velocities. Interestingly, when BRR$> 10$ rpm (Fig. 8 panel
d) small spectral peaks between $4-6$ Hz can be observed at both P2.01 and P3.01, despite the overall low noise amplitude of
the spectra. In order to untangle the possible contribution of the wind farm from other local, wind-related noise sources near
these stations, we show in Fig 9 panel a-b the surface (black curve) and borehole spectra (red curve), together with their 5th,
95th percentiles, for increasing BRR and for wind speed below 5 m s$^{-1}$. At both P2 and P3, while the median borehole spectra

and their percentile intervals remain unchanged as BRR increases, the $4-6$ Hz peaks only appear for the highest BRR. Fig 9
panel c shows the spectra for the same BRR range but for the highest wind speed (a condition that occurred the 9% during the
period of study. Here the borehole spectra detach from the NLNM, and approach the $10^{-9}$ m s$^{-1}$ Hz$^{1/2}$ level, with the spectral
peaks in the $4-6$ Hz range becoming more pronounced if compared to the condition of low wind speed. Moreover, here a $3.4$
Hz peak appears, which is the same signal dominating the PSDs of all stations along the WINES array. These observations

suggest that when the wind park approaches its maximum regime, the seismic noise emitted by the BAS wind park can be
recorded even at the borehole sensors of P2 and P3 ($\sim 250$ m depth), at more than 17 km distance (i.e. the distance between
P3 and the closest turbine). The emergence of such spectral peaks is observed only for the 67% of the observed time, while
for the remaining 33% the PSDs show no effect induced by the nearby wind park. Consequently, on a long-term (i.e. yearly)
time scale the P2 and P3 stations confirm the seismic quietness of the area (Naticchioni et al., 2024) and its adequacy as a

candidate site for hosting ET. In particular, the occurrence of the $4-6$Hz spectral peaks can be explained with two contributing
factors, being i) the exceptionally low level of seismic noise in the study area, which allows the identification of weak seismic
signal produced at distance ii) the rather low damping of the compact, high-velocity, crystalline basement in the area. As an



opposite example, in a site with poorly consolidated sediments and/or with higher background noise, the impact of the wind farm would be hardly detectable at such a distance. Therefore, in the attempt to promote the coexistence of highly sensitive

scientific installations with wind energy production, regulators should be discouraged from applying stringent, fixed-radius buffer zones, as these could be too cautionary for some areas but inadequate for others. In-depth geophysical studies for the characterization of the local geology, noise monitoring, and numerical modelling of the emitted wavefield should be carried out, leading to the estimation *ad-hoc* zone of respect for the specific scientific instrumentation to be placed or already installed. Another strategy for minimizing the impact of wind energy production on sensitive instruments would consist primarily in the

reduction of the emitted seismic noise. For perspective wind farms, physical barriers (i.e. trenches) could be placed around the turbines, confining the noise wavefield through multiple reflections (Abreu et al., 2022). Moreover, mass dampers could be installed (Zhang et al., 2019), and/or piezoelectric materials (Awada et al., 2021) could be used for counteracting the induced vibration. For wind farms already in place, strategies based on the active controls of blade angle, rotation rate, and nacelle tilt can be employed to reduce the produced seismic emission (Bertagnolios et al., 2023). Lastly, a promising strategy might

rely on the real-time, precise tuning of all the operational parameters (Calderaro et al., 2007; Kipchirchir et al., 2022) at each turbine (BRR, tilt and roll angles of the nacelle, as well as the yaw, tilt, and pitch of the blades), such that an out of phase noise could be produced at each turbine, resulting in effective damping of the resulting wavefield by destructive interference.

## 4.2    Limitation and future work

The spectral content, its spatiotemporal variation and its correlation with the operational activity of the BAS wind park are

solid evidences that it is the sole contributor to the seismic noise recorded in the study area. This is further supported by the absence of any relevant anthropogenic noise source that could otherwise explain the observed data. In our analysis, we mainly concentrate on the vertical component of the noise source, on the assumption that most of the emitted noise propagates in the form of Rayleigh waves. This assumption might not be completely valid, since a portion of the emitted wavefield might travel as Love waves. The prediction of energy partitioning into Rayleigh and Love waves remains challenging as it depends on i)

considered frequency and the relative position of the recording station with respect to each turbine and to the instantaneous wind direction (Neuffer et al., 2021). A major source of uncertainty in the estimation of the damping factor $\alpha$ originates from the assumption that each wind turbines act as a simultaneous quasi-random noise source generating the same noise amplitude. In our case the turbines share all the same technical characteristics, therefore this assumption may be regarded as valid at first order. However, second-order effects on the estimation of amplitude decay can arise from a heterogeneous noise emission due

to different soil-turbine coupling and variability of BRR across the wind park. In general, we suggest that more effort should be devoted to evaluating uncertainties when retrieving the damping factor $\alpha$. In fact, while $\alpha$ is solely dependent on the local site condition, its estimation could be biased by the spatial arrangement of turbines, topography, geology, and the phase and amplitude variability of the noise emitted by each turbine across the wind farm. While most of the noise recorded at P2 and P3 can be related to the activity of the wind park, the possible role of wind and other sources excited by the wind could be further

assessed. In this study, we could only rely on a single meteorological station, at more than 20 km from the ET candidate site.





Instead, the collection of dedicated meteorological data at P2 and P3 would be beneficial, helping to decouple the contribution of the wind farm from the local wind conditions.

## 5    Conclusions

A passive seismic experiment was carried out near the "Buddusò-Ala dei Sardi" (BAS) wind park, to characterize the emitted
noise and its possible impact on the noise budget of the nearby candidate site of ET, the third-generation gravitational wave detector. The analysis of the data, retrieved from a 13 km-long linear array of nine broadband stations, led to the following observations:

1. Given the remoteness and seismic quietness of the study area, the BAS wind park can be regarded as the sole contributor to the recorded noise in the frequency band above 1 Hz. This is substantiated by the analysis of Rayleigh wave polar-
ization, which a direction of the incoming seismic noise which is fully compatible with the relative position of the wind park with respect to the array. Signal coherence lessens at greater distances (from WP5 to WP9), as a consequence of the complex noise superposition from multiple turbines.

2. The spectrograms and PSDs show that the noise by the BAS wind park is mainly confined in the $1-10$ Hz frequency range and is dominated by several monochromatic signals. The main spectral picks are found at $3.4$, $5.0$, $6.8$, and $9.5$ Hz
and are recognizable, at the closest stations, even for low blade rotation rates (BRR). The strongest one, found at $3.4$ Hz, can be tracked up to 13 km distance from the wind park.

3. While the amplitude of such spectral peaks increases with increasing BRR, their frequency remains unaffected. These are typical characteristics of the noise corresponding to the fundamental and higher modes of vibration (torsion and oscillation) of the tower structure, whose size is compatible with the $1-10$ Hz range.

4. The amplitudes of the spectral peaks at $3.4$, $5.0$, $6.8$, and $9.5$ Hz, tracked along the entire length of the array, led to the estimation of the damping factor $\alpha$, under the assumption of an exponential decay model with distance. $\alpha \sim 2$ is the rather constant value obtained across different ranges of BRR and for different frequencies. This suggests that the geomechanical characteristics of the local terrain cause the low as well as higher frequency to be almost equally damped with distance.

5. We use the data recorded at P2 and P3, two permanent stations equipped with surface and borehole sensors at $\sim 250$ m depth, for evaluating the possible impact of the BAS wind park at the depth of the perspective Einstein Telescope. When BRR$< 10$ rpm, no increase in the noise budget is observed at these stations, with the surface and borehole sensors showing similar spectra, near the Peterson's NLNM. When BRR$> 10$ rpm, the spectra for the surface seismometers slightly increase and reach (at P3) and surpass (at P2) the level of $10^{-9}$ m s$^{-1}$ Hz$^{1/2}$. On the contrary, the borehole
sensors remain unaffected. This suggests that the rock overburden suppresses most of the emitted seismic noise. Small spectral peaks appear at $3.4$ Hz and in the $4-6$ Hz range, for both P2 and P3. We explain their occurrence with a



combination of low damping characteristics of the local terrain with the exceptionally low level of seismic noise at the site. As confirmed by the long-term, yearly averaged PSDs at P2 and P3 (Naticchioni et al., 2024), the presence of the wind park does not cause any disruption of the seismic quietness of the area.

Our work demonstrates the relevance of the characterization of seismic noise for an adequate evaluation of the disturbance of wind farms on sensitive scientific installations that are negatively affected by seismic noise. This observation should provide the regulators with adequate and *ad-hoc* measures for preserving seismic quietness around these infrastructures, depending on the specific conditions of the local terrain and the target sensitivity of the scientific instrumentation.

*Code and data availability.* ObsPy (Beyreuther et al., 2010) was used for the processing of seismic data. Figures were made with PyGMT
(Uieda et al., 2013) and the Python library `matplotlib`. Seismic recordings of the WINES experiment and those from station P2 and P3 are currently unavailable to the public; data access is granted only to members of the ET scientific collaboration.

*Author contributions.* The WINES experiment was conceptualized and carried out by a collaboration between INGV, INFN, UniSS that include all authors, except for GD, CG, MO and IM who are responsible for the conceptualization and methodology. GD is responsible for data analysis, software, validation, visualization and writing the original draft. The remaining authors contributed to the production and
375 availability of dataset employed in this study.

*Competing interests.* The authors have no competing interests.

*Acknowledgements.* This study was conducted within the framework agreement between Istituto Nazionale di Fisica Nucleare (INFN) and Istituto Nazionale di Geofisica e Vulcanologia (INGV). The study was financed by INFN in the frame of the collaboration agreement with INGV "Accordo di Programma per la Conduzione di Studi e Ricerche Finalizzati alla Caratterizzazione Geofisica e Sismica del Sito di
Sos Enattos (NU)", thanks to the Protocollo di Intesa tra Ministero dell'Università e della Ricerca, Regione Autonoma della Sardegna, the Istituto Nazionale di Fisica Nucleare and the University of Studi di Sassari; by the Università degli Studi di Sassari thanks to the "Accordo di Programma tra la Regione Autonoma della Sardegna, Università degli Studi di Sassari, l'Istituto Nazionale di Fisica Nucleare, l'Istituto Nazionale di Geofisica e Vulcanologia, l'Università degli Studi di Cagliari e l'IGEA S.P.A." (project SAR-GRAV, funds FSC 2014-2020, Patto per lo sviluppo della Regione Sardegna), to the Fondo di Ateneo per la ricerca 2019 and 2020, and to Fondazione di
Sardegna, project 2022-2023: CUP-J83C21000060007. G. Diaferia acknowledges the NRRP-MEET (Monitoring Earth's Evolution and Tectonic) project funded by NextGenerationEU . We thank Geopower and Renantis S.p.A for kindly providing all the data on the wind park activity and wind speed, ARPAS Sardegna and Enrico Giuseppe Cadau for sharing metereological data in the early stage of this work. We also thank Luigi Fodde and the operators of the Sos Enattos underground mine for their valuable help in supporting the preparation of the temporary array.



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



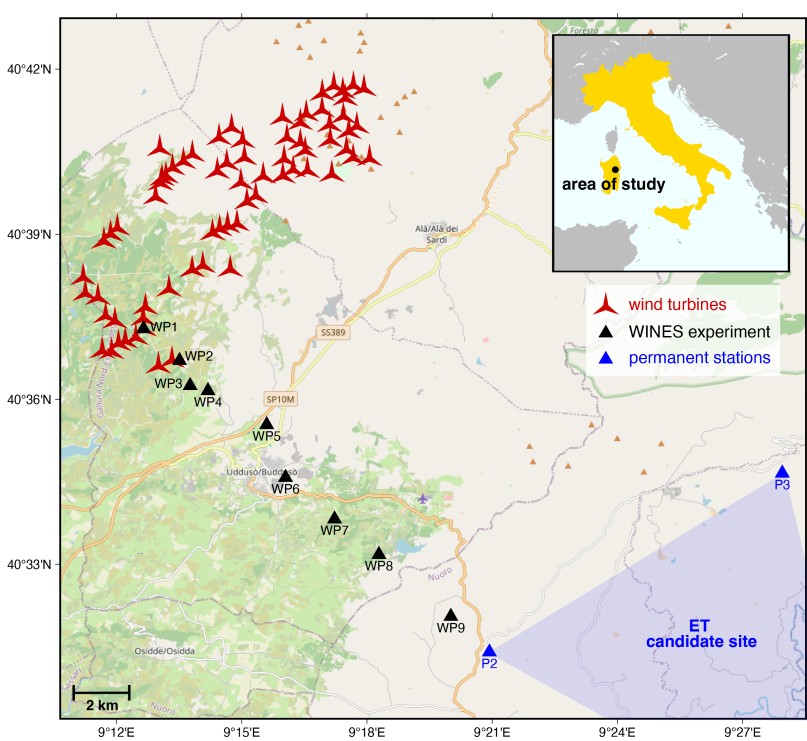

**Figure 1.** Map of the study area. The turbines of the BAS wind park are in red. Black triangles indicates the location of the seismic stations installed within the WINES experiment. P2 and P3 (in blue) indicate the permanent stations located at two of the three vertices of the ET candidate site. Map tile is from ©OpenStreetMap contributors 2024, distributed under the Open Data Commons Open Database License (ODbL) v1.0.



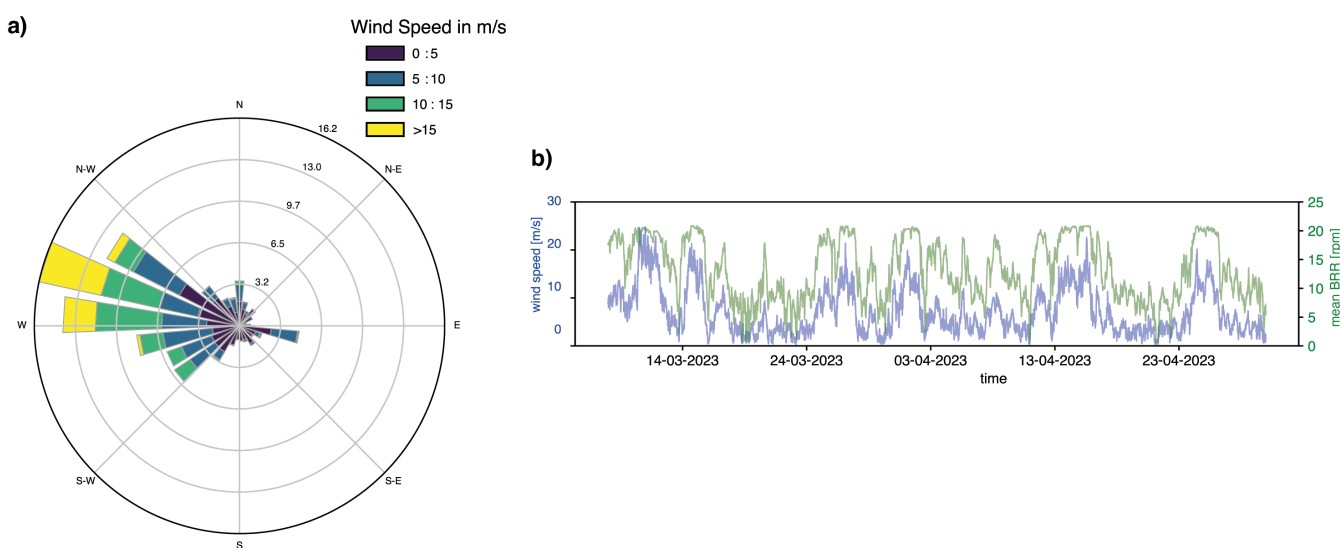

**Figure 2.** a) Histogram of wind velocity and wind direction during the WINES experiment (08/03-30/04/2023), recorded by the 'Met Mast' station, a meteorological located approximately in the center of the BAS wind park. b) Time-series of wind speed (blue) and average BRR (green) across the 69 wind turbines of the BAS wind farm.

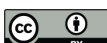



**Figure 3.** a) 10 days long seismic recording (Z component) and spectrogram for stations WP1, WP3 and WP9. b) Time series of wind speed recorded at the 'Met Mast' station.





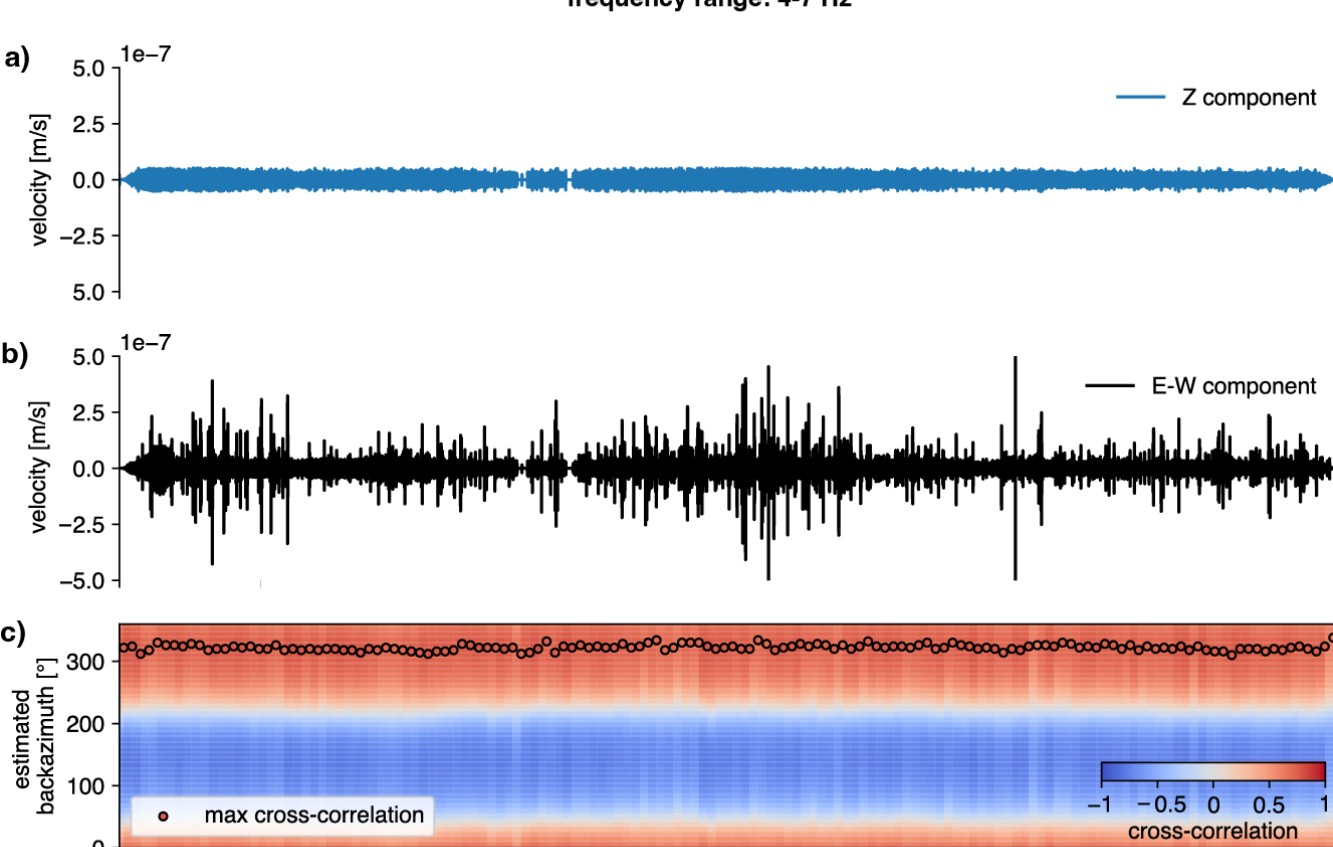

**Figure 4.** Noise direction estimation for station WP3 using the 24 hours long seismic recording of 11/03/2023. In a) and b) the the vertical and E-W component of the seismic record are plotted. c) Estimation of the backazimuth of the incoming seismic noise over 10-minutes long data chunks, based on the cross-correlation of the vertical component and the Hilbert-transformed radial component. Time windows are 10 minutes long.





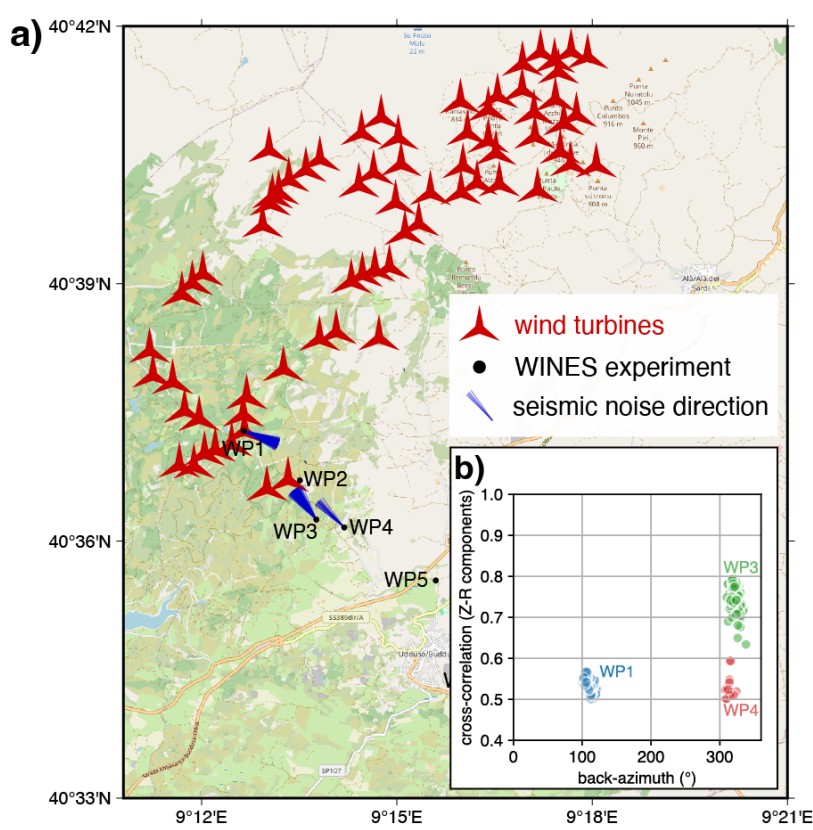

**Figure 5.** a) Map showing the reconstructed backazimuth of the incoming seismic noise (blue lines) at stations WP1, WP3 and WP4. b) Plot of the cross-correlation value vs. backazimuth for stations WP1, WP3 and WP4. Map tile is from ©OpenStreetMap contributors 2024, distributed under the Open Data Commons Open Database License (ODbL) v1.0.



**Figure 6.** Power spectral densities (PSDs) for the seismic stations of the WINES experiment, in addition to the permanent surface sensor (P2.00 and P3.00) bordering the ET candidate site. PSDs are computed for different ranges of BRR across the wind park, ranging from $0-3$ rpm (panel a) to $10-25$ rpm (panel d). The plotted PSD curves correspond to the median value of the PSDs over $10$ minutes long time windows covering the entire duration of the experiment. The dashed lines indicate respectively the Peterson's New High (NHNM) and Low (NLNM) Noise Model (Peterson, 1993).





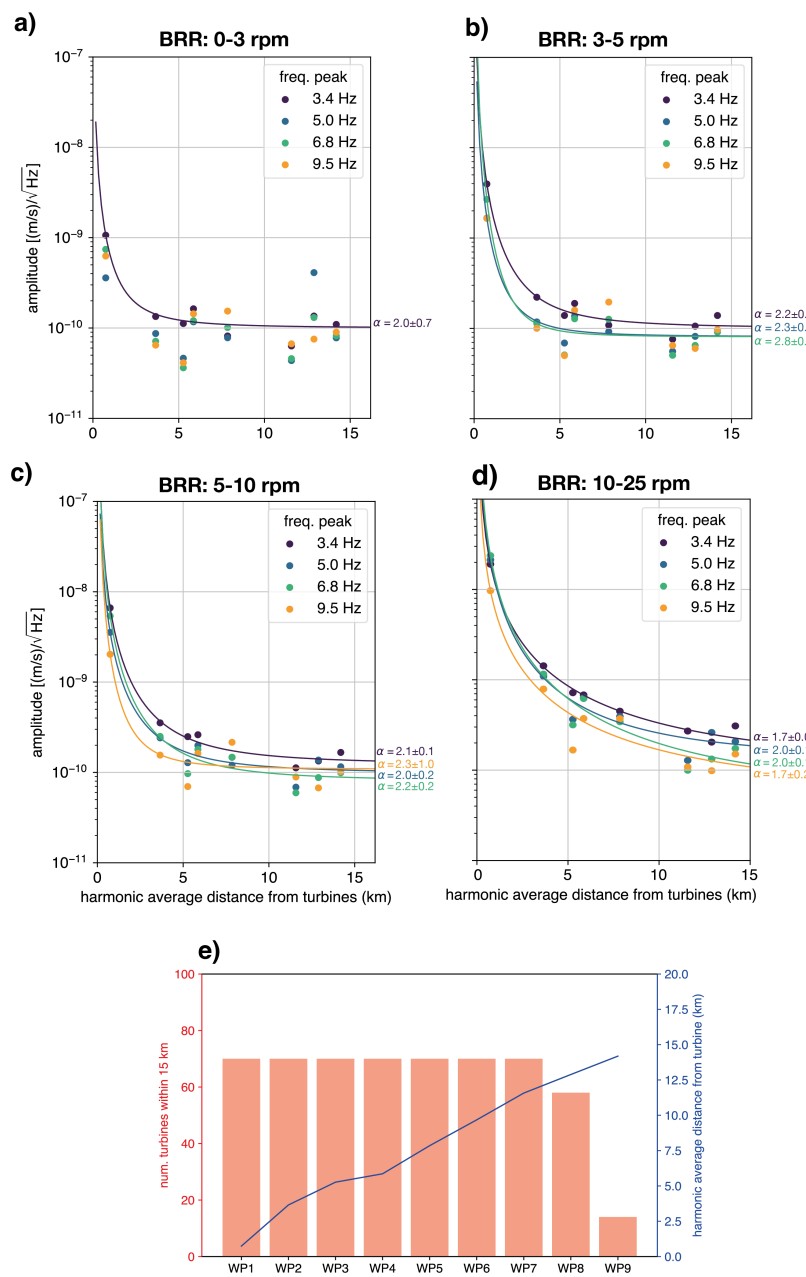

**Figure 7.** a)-c) Decay of spectral amplitudes of the frequency peaks at 3.4, 5.0, 6.8 and 9.5 Hz as a function of harmonic average ($r_H$) from wind turbines, in different range of BRR across the wind park. The fitted curves follow the expression $A \sim r_H^{-\alpha}$. e) In red, the number of wind turbines in a radius of 15 km from each station of the WINES experiment. In blue, the values of harmonic average distance from each station to the turbines. The histogram represents the number of turbines within 15 km from each station.





**Figure 8.** a)-d) PSDs for the permanent sensor at surface (P2.00 and P3.00) and borehole (P2.01 and P3.01) for different ranges of BRR across the wind park. The dashed lines indicate respectively the Peterson's New High (NHNM) and Low (NLNM) Noise Model (Peterson, 1993).





**Figure 9.** PSDs for the permanent stations P2 (a) and P3 (b) for increasing BRR at the BAS wind park and in condition of low wind speed ($< 5$ m s$^{-1}$). c) PSDs for station P2 and P3 in condition of high BRR ($10 - 25$ rpm) and high wind speed ($> 15$ m s$^{-1}$). All PSDs are represented in terms of median value (solid line) and $5 - 95\%$ percentiles. The dashed line indicate the Peterson's New Low (NLNM) Noise Model (Peterson, 1993).