# Peer review of "Seismic noise characterization for the Buddusò - Ala dei Sardi wind park (Sardinia, Italy) and its impact on the Einstein Telescope candidate site"

_EGUsphere, 2024_

## Referee Comment (RC1)

**Review comment for "Seismic noise characterisation for the Budduso-Ala dei Sardi wind park (Sardinia, Italy) and its impact on the Einstein Telescope candidate site, submitted by Diaferia et al.**

This paper evaluates data from a temporary seismic array deployed to study the seismic noise emissions caused by a wind farm in Sardinia, which is located at approximately 20 km distance from the Sardinian candidate site for the Einstein Telescope (ET), a proposed new gravitational wave detector. In addition to the linear WINES array, surface and deep borehole recordings from two permanent seismic stations at the vertices of the proposed telescope area are evaluated.
By analysing spectrograms, power spectral densities (PSDs) and polarization at stations that are close to the wind park, the study identifies several spectral peaks that can be clearly linked to noise from the wind turbines. The behaviour of these peaks with rotation rate of the blades and their decay with distance is then investigated to evaluate the impact of seismic noise from the wind park on the ET candidate site.

The impact of wind noise on observatories is an important question for the Earth Science community and beyond. Therefore, I believe that this manuscript will find many interested readers. I have mostly minor comments, however, I am confused as to the conclusions. It appears to me that the seismic noise generated by the wind park is measurable at the vertices of the ET site, i.e. not absent, and potentially affects a large amount of observation times.

**Major comments:**

The abstract states that "the borehole sensors remain unaffected by the seismic noise[...] small residual spectral peaks at 3.4 Hz and between 4 – 6 Hz remain detectable" → this is contradictory, if the noise is detectable then the borehole sensors are not unaffected. See also lines 290 ff, which, if I understand well, suggest that the high-rotation rate conditions could prevail in 2/3 of cases; see also conclusion 5.

In terms of the 1/sqrt(N) correction for the amplitude, I checked the reference to Schofield (2001) and they use this with the intention of modeling the amplitude at locations where data are not available. What I find a bit problematic with regard to how it is used here and in previous studies is that the cutoff distance of "visible turbines" is somewhat arbitrary, here 15 km are chosen, while in another study 10 km are chosen and so on. To make damping exponents more easily comparable, I suggest to include e.g. in the supplement the results for the damping without the 1/sqrt(N) correction (i.e. directly comparable to Zieger's results and more easily comparable to other results without choosing a distance threshold).

There are sharp, seemingly quasi-monochromatic peaks in the noise spectra e.g. at P3 between 2 and 3 Hz, or at both P2 and P3 between 8 and 9 Hz and between 9 and 10 Hz. Given that the study is preoccupied with the seismic noise at the site, I wish these were also described and discussed, and eventually included in conclusion 1. The peak between 8 and 9 Hz, for example, appears to be visible at multiple stations and could be related to another source of anthropogenic noise.

**Minor comments:**
- there are several unopened or unclosed parenteses; I hope that the typesetting will spot these, e.g. line 151
- several references are missing the parentheses, e.g. line 121, line 126, line 238
- line 45: instead of an inline URL citation, I suggest to include a proper URL reference in the reference list with last accessed date
- line 124, aerially scattered – I was not sure if this means scattered in the air, or scattered in an area (areally?), please clarify for the readers
- line 127 "divide by N^(-1/2)" should be "divide by N^(1/2)"?
- line 190 For the case of BRR ... this sentence may be missing a verb
- line 200 "panel d)" → should be panel e?

---

## Author Response (AR1)

**Rebuttal Letter**

**Reviewer #1**

Major comments:

The abstract states that "the borehole sensors remain unaffected by the seismic noise[...] small residual spectral peaks at 3.4 Hz and between 4 – 6 Hz remain detectable" → this is contradictory, if the noise is detectable then the borehole sensors are not unaffected. See also lines 290 ff, which, if I understand well, suggest that the high-rotation rate conditions could prevail in 2/3 of cases; see also conclusion 5.

We recognize that these statements are contradictory and thank the reviewer to have pointed this out. These have been rephrased and clarified.

In terms of the 1/sqrt(N) correction for the amplitude, I checked the reference to Schofield (2001) and they use this with the intention of modeling the amplitude at locations where data are not available. What I find a bit problematic with regard to how it is used here and in previous studies is that the cutoff distance of "visible turbines" is somewhat arbitrary, here 15 km are chosen, while in another study 10 km are chosen and so on. To make damping exponents more easily comparable, I suggest to include e.g. in the supplement the results for the damping without the 1/sqrt(N) correction (i.e. directly comparable to Zieger's results and more easily comparable to other results without choosing a distance threshold).

Following the reviewer suggestion, we show in the SM the plot of the PSD decay without the $N^{1/2}$ scaling. It is interesting to note that, while fitted amplitudes are now different compared to those in Fig. 7 of the manuscript, the inferred decay law for each PSD peak does not show an appreciable change. This is likely due to the clustered, rather than scattered, arrangement of the wind turbine with respect to the seismic array.

There are sharp, seemingly quasi-monochromatic peaks in the noise spectra e.g. at P3 between 2 and 3 Hz, or at both P2 and P3 between 8 and 9 Hz and between 9 and 10 Hz. Given that the study is preoccupied with the seismic noise at the site, I wish these were also described and discussed, and eventually included in conclusion 1. The peak between 8 and 9 Hz, for example, appears to be visible at multiple stations and could be related to another source of anthropogenic noise.

The reviewer points out an interesting feature that was not discussed in the manuscript. We now explain in the body of the manuscript that these quasi-monochromatic peaks at P2 and P3, are likely not related to the wind park. In fact, while the whole PSD curves show at least a very small shift for increasing BRR of the wind park, these quasi-monochromatic peaks remain unchanged. Given the remoteness of the area and the lack of any appreciable anthropic activity and infrastructure (e.g. railroad, industries, main roads, quarry, large cities) in the vicinity (<10 km), we suppose these are remnants of low amplitude anthropogenic noise generated at large distance, which becomes detectable at the site due to its quietness and the low seismic damping of the local terrain.

Minor comments:

- there are several unopened or unclosed parenteses; I hope that the typesetting will spot these, e.g. line 151 **CORRECTED**
- several references are missing the parentheses, e.g. line 121, line 126, line 238 **CORRECTED**
- line 45: instead of an inline URL citation, I suggest to include a proper URL reference in the reference list with last accessed date **CORRECTED**
- line 124, aerially scattered – I was not sure if this means scattered in the air, or scattered in an area (areally?), please clarify for the readers **CORRECTED**
- line 127 "divide by $N^{(-1/2)}$" should be "divide by $N^{(1/2)}$"? **CORRECTED**
- line 190 For the case of BRR ... this sentence may be missing a verb **CORRECTED**
- line 200 "panel d)" → should be panel e? **CORRECTED**

**Reviewer #2**

The paper investigates the noise generated by wind turbines (WT) of a wind park and its influenceon the planned Einstein telescope, a seismically sensitive installation. The noise decay has beenobserved using a temporary seismic station set which also verified a small noise contributionon permanent seismic borehole stations in the area of the Einstein telescope. The methods appliedhave been described in a number of publications before.

I 19: I would prefer a formulation "... indicating *a* significant noise suppression at depth.". There are examples for borehole locations where almost no noise suppression can be observed (e.g. Stations IU.GRFO/GR.GRA1 or GR.GOR5). The noise suppression depends on frequency (as mentioned in the text later) and (mainly) whether or not the borehole reaches another possibly noise reflecting geological layer. [R1]
We corrected the next as suggested, now specifying that the noise suppression occurs in the specific frequency range of interest (1-10 Hz)

I 44: comment: these turbines are more or less tiny compared to modern on-land turbines with atotal height between 200 and 300m and installed power of 7MW. [R2]
We now specify that, despite the moderate height of the turbines, the BAS wind park remains among the largest in Italy.
74: How were the stations installed? Was WP1 in particular protected against infrasound (buried)? [R3]
We now specify that all sensors were buried underneath the soil surface.

I 84: abbreviation BRR not used here [R4]
Corrected
I 86: at which height the wind speed data were measured? [R5]
Wind is measured at a height of 64 m. Now this is inserted in the text.

l 92: Only three wind speed intervals listed, 15-25 m/s missing. [R6]

Corrected

l 111: the PSD is computed on time series, i.e. no continuous integration applicable. Which digitalalgorithm is used (library?) [R7]

The algorithm is from McNamara & Buland [2004]. This is now cited in the text.

l 150: The figure suggests that station WP9 itself is sensitive to local wind. This is unfortunateas this will overprint possible contributions of the WT generated noise. [R8]

We cite that this station is probably poorly shielded from the wind action.

l 174: Does BRR or wind speed show a better correlation with the amplitudes of WT generated noise? In line 96 it is stated that already low wind speeds generate a high BRR. I would expect that the noise generation scales with the forces acting on the WT (i.e the wind load), the BRR contribution is added but saturates early. Therefore, why the spectra are binned over BRRand not over wind speed? [R9]

The choice of BRR vs. wind speed has been a matter of discussion during this study. Initially, our analysis was done relying of wind speed data, provided from an amateur meteorological station in the vicinity of the wind park. Later the data on the wind park operation became available, and we decided to bin our data based on BRR, as it is a better proxy on the activity of the wind park. In fact, by using wind speed binning, it is difficult to decouple the effects related to the wind farm and those locally induced by the wind (see the case of station WP9). A convincing argument is provided in the Discussion section (see Fig. 9): at P2 and P3 we see that in conditions of  high BRR and strong wind we see a general increase of the noise level compared to the case of only high BRR and low wind, testifying that the wind act as an additional and local source of noise at each station.

l 176: Can you exclude that there is a considerable effect of direct sound interaction with the seismometer at WP1? Sound would impose BRR induced frequencies and their multipleson the spectrum. [R3]

Given the close distance of WP1 to the turbine, a possible contamination by infrasound emission cannot be excluded. However, whether and how infrasounds can reach a buried seismogram through the air-to-ground conversion of the pressure wave is challenging to assess, and depends on the soil density and its elastic properties. Following the work from Gortsas et al. (2017) based on numerical modelling for the investigation the role of seismic and infrasound energy radiated by a wind turbine, we conclude that most of the energy propagates as Rayleigh waves whose disturbance is larger than the converted air-borne infrasound. In terms of recovery of the amplitude decay law, we exclude that the contamination of infrasound noise at WP1 introduces any considerable bias. In fact, we use the amplitude of the 4 strongest peaks in the PSD (related to the turbine vibration and recognizable at all stations), which are for sure related to the Rayleigh wave seismic noise emission (see also the noise direction analysis, see Fig. 5).

As an exercise, we tried to exclude the amplitude of WP1 from the set of points to be fitted to retrieve the damping factor $\alpha$. We observe that the exponentiality of amplitude vs. distance is appreciable only for BRR 10-25 rpm and the retrieved is between 2.4 and 2.6, a range which is within the error on the estimation of $\alpha$ across different frequencies and BRR (See Fig. 1)

[Figure]

Fig. 1 Plot of amplitude vs. distance fit. Amplitude data for station WP1 are shown but not used for the fit.

304: To my knowledge there are so far no seismic measurements to verify these theoreticallysuggested mitigation strategies in real life. A stringent comparison experiment wouldbe very expensive as it would mean to build isolated WTs with and without mitigationdevices next to each other and measure the difference in emissions. Additionally, at least someof the methods (artificial trenches) seem pretty impracticable to realize. In my opinion thisdiscussion is purely academic for the time being. [R10]

We thank the reviewer for the comment. To the original text we add that these mitigation measures are somewhat speculative and/or at an early state of development with no assurance of their employability in the near future.